# Experimental Study at the Reservoir Head of Run-of-River Hydropower Plants in Gravel Bed Rivers. Part II: Effects of Reservoir Flushing on Delta Degradation

**Kevin Reiterer \*, Thomas Gold, Helmut Habersack, Christoph Hauer and Christine Sindelar**

CD-Laboratory for Sediment Research and Management, Institute for Hydraulic Engineering and River Research, Department for Water—Atmosphere—Environment, BOKU—University of Natural Resources and Life Sciences Vienna, 1190 Vienna, Austria; thomas.gold@boku.ac.at (T.G.); helmut.habersack@boku.ac.at (H.H.); christoph.hauer@boku.ac.at (C.H.); christine.sindelar@boku.ac.at (C.S.)

\* Correspondence: kevin.reiterer@boku.ac.at

**Abstract:** Run-of-river hydropower plants (RoR HPPs) are capable of interrupting the sediment connectivity of many alpine rivers. Still, there is a lack of systematical investigations of possible sediment management strategies for small and medium sized RoR HPPs. This study deals with the headwater section of an impoundment and the approach of sediment remobilization during drawdown operations. Therefore, a typical medium sized gravel bed river having a width of 20 m, a mean bed slope of 0.005, a mean flow rate of 22 $m^3$/s, and a 1-year flood flow of 104 $m^3$/s is recreated by a 1:20 scaled physical model. Heterogenous sediment mixtures were used under mobile-bed conditions, representing a range of 14–120 mm in nature. During the experiments, the flow rate was set to be 70% of the 1-year flood ($HQ_1$) regarding on the ability to mobilize all sediment fractions. The possibility to remobilize delta depositions by (partial) drawdown flushing within a reasonable period ($\approx$9 h in 1:1 scale) was shown by the results. The erosion of existing headwater delta deposition was found to be retrogressive and twice as fast as the preceding delta formation process. A spatiotemporal erosion scheme points out these findings. This supports the strategy of a reservoir drawdown at flood events of high reoccurrence rate.

**Keywords:** gravel bed rivers; run-of-river hydropower plants; sediment continuity; flushing events; delta formation; sediment management strategies; retrogressive erosion

## 1. Introduction

Hydropower constitutes the historically most widely used renewable energy source. All Alpine countries in Europe started the production of electrical energy with run-of-river hydropower plants (RoR HPPs) in the same period around 1880 [1]. After the first world war (WW I), the importance of hydropower increased. With the end of the second world war (WW II), HPP construction boomed, lasting till the 1980s. During the 1970s, a broad public discussion about negative ecological effects caused by hydropower began, initiating a turning point. This led to a decreasing number of new HPPs [1]. With the implementation of the Water Framework Directive (WFD) [2] in 2000 and the aim to establish the good ecological status in European freshwater systems, hydropower faced new challenges. This affected new projects as well as operating HPPs [1]. Still, hydropower is capable of playing an important role in the energy transition [3]. The Revised Renewable Energy Directive RRED [4] of the EU clarifies its importance.

The Alps are often described as Europe's water reservoir and constitute an important sediment source for big streams like Rhone, Rhine, Po, and Danube [5]. In the Alpine region, defined by the Alpine Space programme [6], 1019 HPPs with a capacity >5 MW (RoR-HPP > 5 MW) are situated. These subdivide into 59% RoR HPPs, 33% storage plants, and 8% pumped storage plants. Together they contribute about 40% to the total hydropower capacity of the European Union [7]. In Austria 115 RoR-HPP > 5 MW are in operation. Their share is about 4% of all 2770 RoR HPPs located in Austria and only 3% exceed a capacity of 10 MW [8]. Despite the capacity or size, every single RoR HPP is capable of disrupting the natural river continuum and therefore matters in terms of sediment connectivity, which is one key aspect to reach a "good ecological status" as described in the WFD [2].

The focus of this paper was to investigate sediment transport processes at low head RoR HPPs located in medium-sized gravel bed rivers, since most of the existing literature is about large dams, with large storage volumes, dominated by different sedimentation characteristics [9]. During projected operations at RoR HPP facilities, usually the incoming bed load material is held back in the impoundment, while most parts of the suspended load is expected to pass the dam [10]. Starting at the headwater section of an impoundment, the flow parameters rapidly change due to the reduction in energy slope. The flow depth increases continuously while the flow velocity and the sediment transport capacity decrease. This causes deposition and sorting of the incoming material. Due to the aggregation of coarse particles at the head of the reservoir, typical delta formations occur. Finer bed load fractions are carried further downstream, causing reservoir sedimentation [11]. Therefore, sustainable sediment management strategies are necessary to extend the life expectancy of the RoR impoundment. A disrupted sediment balance further impacts the downstream river reach, leading to uncontrolled bed erosion and land loss [12]. There is a rising awareness of the importance of sediment management strategies among HPP operators initiated by changing legal, political, social, and scientific conditions [9]. Still, there is a deficit in systematical approaches how sediment management strategies at RoR HPPs must be established. According to Kondolf et al. [13] reservoir sediment management strategies split up into three main categories: (i) reduction of the sediment yield from the catchment area, (ii) reduction of the sediment deposition, and (iii) volume increase and recovery. Least anthropogenically disturbed sediment connectivity constitutes the best ecological state and should therefore be reflected in the choice of management strategies. Per definition sluicing and flushing differ by two main aspects. First, sluicing aims at the passing of sediments through the reservoir without deposition, while flushing refers to the removal of preliminary deposited sediments. Secondly, flushing is not necessarily connected to the occurrence of high flow rates. However, non-flood flushing is assumed to only be suitable for reservoirs with large storage volumes and low-level outlets [13,14]. Hence, for small and medium sized RoR HPPs sluicing, as well as flushing operations, depend on high flow rates. According to [14] reservoir flushing is often dominated by backward evolving erosion. This process, termed as "retrogressive erosion", is often caused by discontinuous longitudinal profiles which lead to a change in hydraulic energy. Retrogressive erosion is indicated by high erosion rates and an upward moving degradation process [14].

The present paper constitutes Part II of a two-part study concerning sediment management of RoR HPPs in medium sized gravel bed rivers. Part I deals with delta formation processes at operation water levels and forms a prerequisite of this paper [9]. The aim of this present Part II is (i) to investigate the reservoir sedimentation of a low-head RoR HPP, (ii) to characterize the erosion process of existing headwater delta aggregations during low flood drawdown flushing, and (iii) to discuss the results in terms of future HPP operation. The discharge during the experiments was set to 70% of the 1-year flood ($0.7 \times HQ_1$), since this constitutes a good standard for initiating drawdown operations [9,15].

## 2. Experimental Setup and Methods

### 2.1. Hydrological Characteristics of the Medium Sized Gravel Bed River

The hydrological model conditions are defined by the Mürz river in Styria, Austria. The Mürz river is a representative medium sized gravel bed river of the Alpine region, located in the northern Central Alps. Along its flow path of 83 km the Mürz river hosts 23 RoR-HPPs and six diversion plants [16]. A detailed hydrological analysis of the gauging station Kapfenberg-Diemlach (480 m a.s.l.), which is located 2.3 km upstream of the river's entry into the Mur, was carried out. The hydrological data was provided by the Austrian Hydrographic Service. Table 1 sums up significant hydrological data. For the period between 1971 and 2016 the mean monthly flow rates were calculated based on the average daily flow rates. Further, the daily flow rates were subdivided into six discharge ranges and the corresponding number of the events was computed. This information is used for a more detailed analysis of annual flood patterns.

**Table 1.** River parameters and hydrological characteristics at the gauging station Kapfenberg-Diemlach; MQ = mean flow, $Q_d$ = design flow, $HQ_1$ = 1-year flood, $HQ_5$ = 5-year flood, $HQ_{10}$ = 10-year flood. The x- year flood ($HQ_x$) is a flood event which has a chance of 1/x of being reached or exceeded in any given year.

| River Parameters | Flow Rate ($m^3$/s) |
|---|---|
| Catchment area 1364.5 $km^2$ | MQ = 22 |
| River width $W_0$ = 20 m | $Q_d$ = 35 |
| River length = 83 km | $HQ_1$ = 104 |
| Bed slope $S_0$ = 0.005 | $HQ_5$ = 150 |
| Mean grain diameter $d_m$ = 0.056 m | $HQ_{10}$ = 180 |

### 2.2. Annual Reservoir Sedimentation

By fitting a tri-linear regression model to the measured sediment transport rates of the free-flowing section provided by [9], a simplified transport model was computed and is presented in 1:1 scale. Sindelar et al. [9] compared the measured transport rates with five different well-known transport formulas. The deviation between the measurements and four of the transport functions is within +/−25%. When combining the transport model with the average daily flow rates the mean annual sediment transport for the period between 1971 and 2016 was calculated. Based on the frequency distribution for different flood events the share of the total bed load transport for each discharge range was determined. The calculations assume full sediment availability in the free-flowing section, i.e., sediment transport is transport limited rather than supply-limited. Together with the knowledge of the initial reservoir volume C and the annual water inflow I the monthly reservoir sedimentation rates were derived. The monthly rates sum up to the annual reservoir sedimentation rate (RSR). Based on these calculations a comparison to field data at 1:1 scale was made to discuss the quality of the physical modelling.

### 2.3. Experimental Setup

The physical model was constructed in the Hydraulic Laboratory of the University of Natural Resources and Life Science Vienna to investigate sediment transport processes. The 1:20 scaled model is built up of three different sections, as displayed in Figure 1a: (i) an inlet section containing flow straighteners, (ii) a 9.5 m long and 1 m wide experimental section with mobile bed conditions, and (iii) an outlet section. The model inflow is regulated with a magnetic flow meter, situated at the pipe inflow. At the end of the inlet basin a sediment feeder was used to constantly add the amount of required sediments. The technical details of the sediment feeding machine are described in [9]. The outlet section consists of a sediment trap, which is used for the collection and the underwater weighing

of the entrained sediments as well as the flap gate to adjust the water levels during the experiment. According to the law of Froude similarity a length scale of 1:20 leads to a time scale of $1 : \sqrt{20}$.

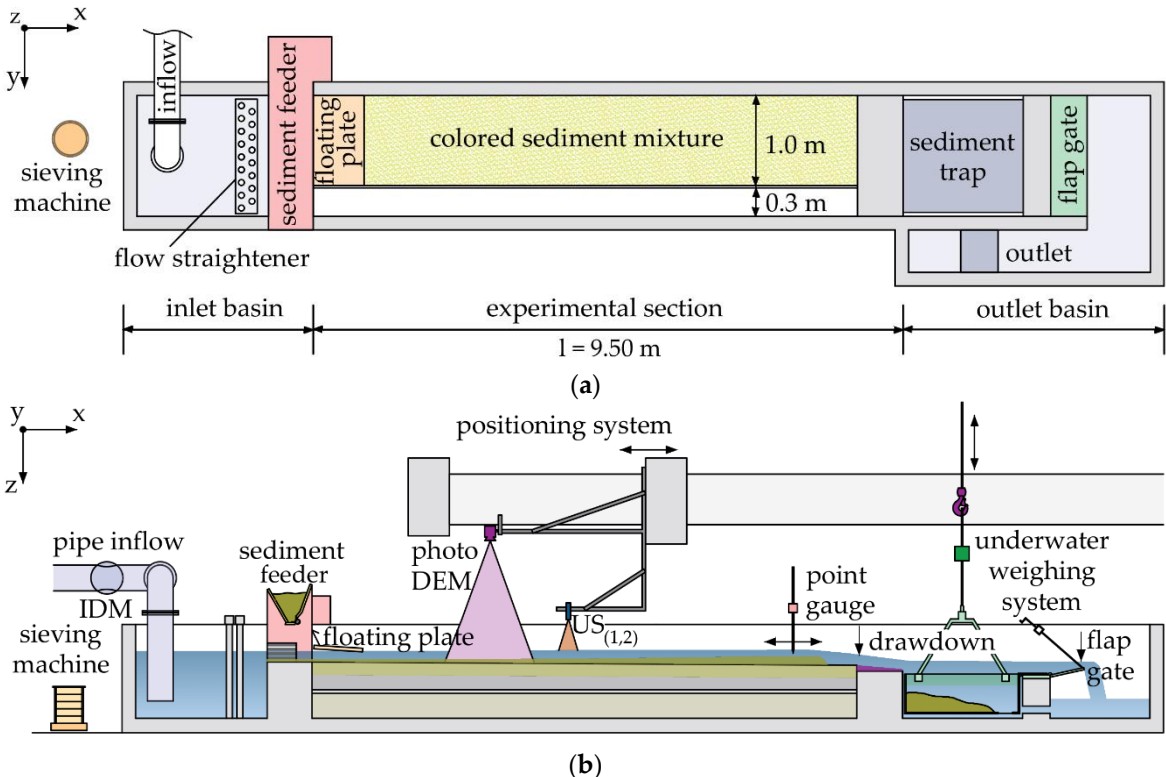

**Figure 1.** Experimental setup, (**a**) plan view; (**b**) longitudinal section of the straight flume. DEM = digital elevation model, US = ultrasonic probe.

Figure 1b shows the longitudinal profile of the experimental setup. During the test runs the water surface levels were measured by means of ultrasonic probes. In addition, a point gauge was used for check measurements. The evolution of the bed levels was recorded via photogrammetry with a measuring interval of 2 h. For this reason, ground control points were installed on both sides of the experimental section. After carefully draining the model 45 pictures (6000 × 4000 pixel) were taken using a Nikon D7100 (Nikon, Tokyo, Japan) with an AF NIKKOR 20 mm lens (photo DEM). The pictures overlapped by approximately 80%. The software Agisoft Metashape 1.5.2. (Agisoft LLC, St. Petersburg, Russia) was used for processing the pictures. They were aligned and a digital elevation model (DEM) with a resolution of 0.012 × 0.012 m was produced.

The sediment mixture used in the scaled model comprised grain sizes between 0.7 and 6.0 mm, representing a range from 14 to 120 mm at 1:1 scale. The material has a density of 2600 kg/m$^3$ and a bulk density of 1700 kg/m$^3$. Regarding the comparability of sediment transport in scaled models and in nature, process similarity must be guaranteed. This was done by ensuring the same Shields parameter θ in the 1:20 and 1:1 scale. The similarity of Re * may be neglected, if a threshold value of Re* >100 is exceeded [9,17]. This was true for the conducted experiment, as the sediment and the fluid density scale was 1 and a value for Re* of >150 was reached during all tests of this study.

### 2.4. Experimental Procedure

The physical model represented the head of a reservoir of a RoR HPP in a straight section of the Mürz river, having a length of 170 m in 1:1 scale (marked as experimental section in Figure 2). Part I of Sindelar et al. [9] contained two major test series: (i) the examination of sediment transport capacities for a free-flowing section and (ii) the determination of sediment transport processes during turbine operation at the head of a reservoir. The initial conditions for the model tests of the present

Part II were provided by the results of the delta formation experiments of [9]. The delta formed in these experiments was further used as the initial bed surface level. The physical model tests included two separate test runs based on an existing delta formation. Before starting the actual flushing experiments, the water surface level (WSL) was kept at operation level of the hydropower plant. By lowering the flap gate at the downstream end of the model a drawdown was simulated, as shown in Figure 1b. In general, one can distinguish between (i) a full drawdown, which leads to free-flowing conditions across the entire impoundment and (ii) a partial drawdown with remaining backwater effects in the vicinity of the weir. Due to the partial drawdown, shown in Figure 2, the initial head of the reservoir moves towards the HPP resulting in free-flowing conditions for the experimental section. This states the beginning of the experiment ($t = 0$). Hence, the change in the flow conditions is expected to cause an erosion of the existing delta. To capture the ongoing processes WSL measurements were carried out at several time steps. As the erosion process is assumed to be nonlinear the WSL were measured more frequently during the first 2 h of the experiment. Besides the WSL the corresponding bed surface levels (BSL) were further recorded every 2 h to detect possible trends in the WSL and BSL alterations. The total time length of each test run added up to 8 h. Further on, the entrained material was measured every hour by hydrostatic weighing. The amount of fed sediment was set to a constant value (52 kg/hm) based on the subsequent reference test runs for a free-flowing section, also described in [9]. Figure 2 shows in a conceptual way a longitudinal section of a RoR-HPP during a flushing event.

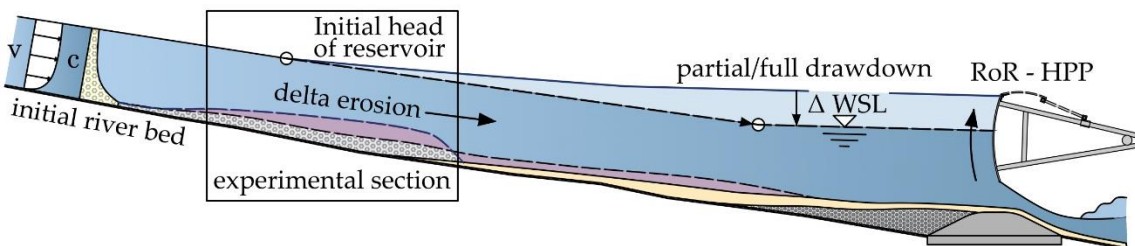

**Figure 2.** Section of a run-of-river hydropower plant (RoR-HPP) reservoir during a flood event and a partial drawdown, leading to free-flowing conditions in the experimental section.

### 2.5. Volumetric Analysis of the Existing Delta Formation and the Ensuing Flushing and Erosion Process

With the photogrammetric data analysis, alterations in the delta aggregation as well as the spatiotemporal processes of the delta erosion were investigated along the flow path. Starting from the longitudinal station X = 2.5 m, the experimental section was divided in six sections ($S_i$), each with 1 m length in x-direction and a width of 0.7 m in y-direction. The first 2.5 m of the experimental section were not considered in the analysis due to a possible influence of the inflow section (boundary influence). Figures A1 and A2 in the Appendix A section illustrate this procedure. By subtracting the digital elevation models (DEMs) at different time steps the elevation changes were computed as a raster dataset. By multiplying the mean height change per section ($\Delta h_i$) by the section area ($S_i$) the volume change ($\Delta V_i$) was computed for every section ($n = 6$) and all the time steps ($n = 4$). Those volume changes ($\Delta V_i$) were assigned to the centroid of its corresponding section ($S_i$). These steps were done for the preliminary delta formation and the delta erosion during the flushing experiments. For the delta formation in [9] the DEM after 8 h of experiments was subtracted by the DEM of its initial riverbed. For the flushing experiments, the DEMs after 2, 4, 6, and 8 h were each subtracted from the final delta formation DEM after 8 h, therefore the changes volumes refer to the same starting point. For the gathered data set a statistical trend analysis was carried out. To make the different test runs comparable, the volume changes ($\Delta V_i$) were normalized by following Equation (1):

$$\Delta VS_i = \frac{\Delta h_i \times S_i}{\Delta h_{mean} \times S_i} \tag{1}$$

In Equation (1) $\Delta h_i$, mean refers to the mean height change of all sections. This leads to the normalized volume changes ($\Delta VS_i$) in $m^3/m^3$ for each test run. By means of linear regression, possible trends and correlations were detected.

## 3. Results

The results section includes three main parts. The contribution of different flood events to the annual reservoir sedimentation is described in Sections 3.1 and 3.2. In Section 3.3 the results of the flushing experiments are presented, including water and bed surface evolution, sediment transport rates, as well as a detailed analysis regarding the predominant erosion processes. If not otherwise stated, the results of Sections 3.1 and 3.2 are given in 1:1 dimension. The results of Section 3.3 are presented in model dimensions (1:20) as this corresponds to the direct measurements.

### 3.1. Hydrological Analysis

Figure 3a shows the mean monthly flow rates of the gauging station Kapfenberg-Diemlach for the years 1971–2016. The data is based on the average daily flow rates, provided by the Austrian Hydrographic Service. The mean monthly flow rates have their peak value in April with decreasing values for May and June. Hence, this indicates a nivo-pluvial regime [18].

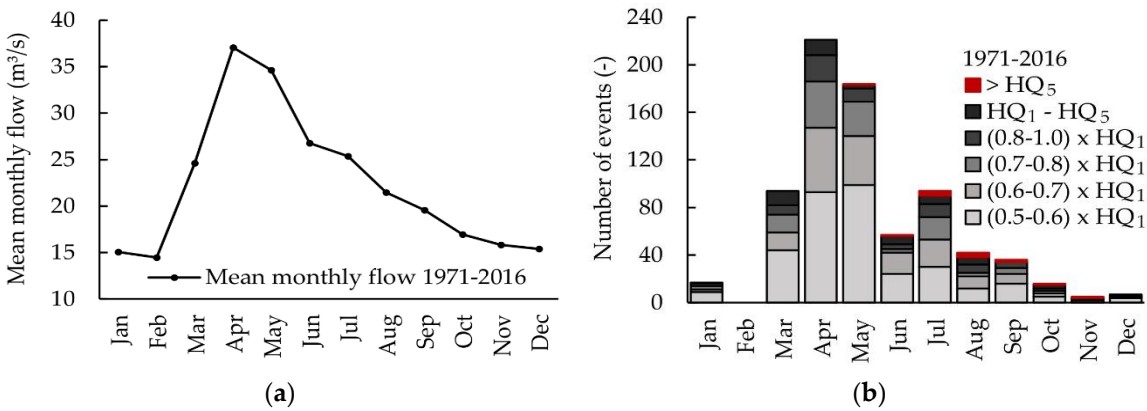

**Figure 3.** Comparison of hydrological data from the Kapfenberg for the years 1971–2016, (**a**) mean monthly flow; (**b**) frequency distribution for different flood events. Hydrograph data retrieved from www.ehyd.gv.at on 10 March 2019.

Figure 3b illustrates the cumulated monthly distribution of flood events above $0.5 \times HQ_1$ for the years 1971–2016. By subdividing the number of events into different discharge ranges, a more detailed analysis regarding annual flood patterns was possible. The flow rate of $0.5 \times HQ_1$ which is 2.3 times larger than the mean flow (MQ) was exceeded 773 times in the observed period of 45 years. This corresponds to about 5% of the total number of days. A flow rate in the range of $0.5$–$0.6 \times HQ_1$ occurred at 338 days. In total, 70% of these events took place in March, April, and May, respectively. In June, the total number of events $>0.5 \times HQ_1$ is clearly smaller than in July, although Figure 3a shows a higher mean monthly flow rate. This indicates the switch from the nival to the pluvial dominated flow period. Flow rates above $0.6 \times HQ_1$ only occurred in 2.6% of the total number of days. The mean annual occurrence of flood events $>0.7 \times HQ_1$ was found to be six for the Kapfenberg-Diemlach gauging station. For flow rates $>HQ_5$ the months July and August show the highest number of events (four each), followed by September (2). This means that 71% of the flood events $>HQ_5$ appeared in summer. Another interesting observation is the lack of events $>0.5 \times HQ_1$ in February. The knowledge regarding distribution and corresponding magnitude of flood events is crucial for adapting sediment management strategies.

### 3.2. Annual Reservoir Sedimentation

Figure 4a shows the transport rates for a free-flowing section for different flow rates together with the fitted tri-linear transport model (Section 2.2). The model parameters are shown in Table 2. The discharges range from $0.0 \times HQ_1$ to $0.8 \times HQ_1$. The first part of the model represents the low discharges to $0.25 \times HQ_1$ and therefore small transport rates. The second line covers the values between $0.25 \times HQ_1$ and $0.7 \times HQ_1$, and the third one refers to all values above $0.7 \times HQ_1$.

**Table 2.** Parameters of the tri-linear transport model in 1:1 scale.

| Q-Range | Q-Range (m³/sm) | Slope |
|---|---|---|
| $0–0.25 \times HQ_1$ | 0–2.5 | $3.3 \times 10^{-6}$ |
| $0.25–0.7 \times HQ_1$ | 2.5–3.7 | $6.2 \times 10^{-4}$ |
| $>0.7 \times HQ_1$ | >3.7 | $1.8 \times 10^{-3}$ |

With an estimated reservoir length (lres) of 1500 m, a dam height (hd) of 5 m a reservoir capacity of $C = 85 \times 10^3$ m³ can be calculated. Based on the annual inflow of $I = 7 \times 10^8$ m³ this leads to a C/I ratio of $1.2 \times 10^{-4}$ m³/m³. Further, the mean monthly sedimentation rate (RSR) was computed for different discharge ranges. The result is an annual RSR of about 43%.

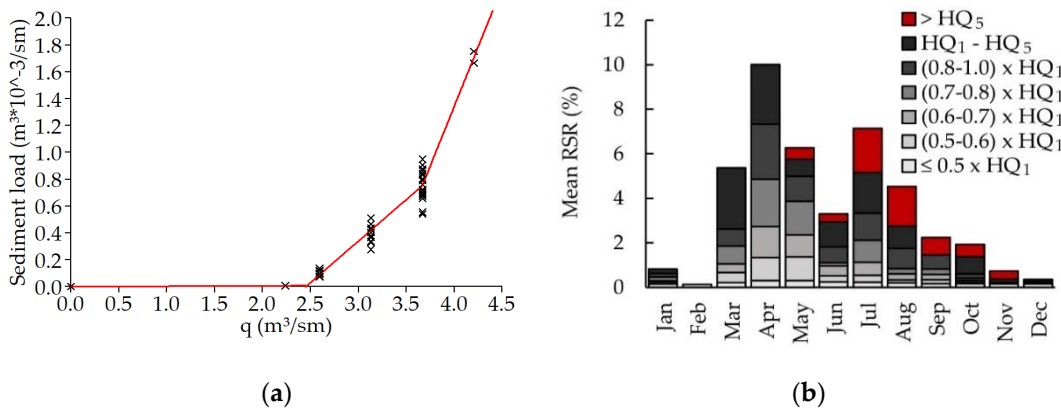

(**a**)　　　　　　　　　　　　　　　　　　　　　　　　(**b**)

**Figure 4.** Incoming sediment load and reservoir sedimentation rate RSR, (**a**) tri-linear sediment load model; (**b**) monthly RSR distribution for different discharge ranges based on (**a**).

Figure 4b shows the resulting monthly shares of the annual RSR. The cumulated distribution of the monthly shares looks similar to the one presented in Figure 3b. However, a reversal of the trend appears regarding the contribution of each discharge range to the monthly RSR. While the events $>0.7 \times HQ1$ constitute about one third of all events $>0.5 \times HQ1$, they contribute 75% to the annual RSR. Although this discharge range shows a mean occurrence of only six days per year, it contributes the most to the reservoir sedimentation, according to the tri-linear sediment model. Splitting this further into discharges from $0.7 \times HQ1$ to HQ1, this refers to 1/4 of all events $>0.5 \times HQ1$ having a share of 1/3 in the RSR. This range shows a mean reoccurrence of four days per year. This points out the eligibility of $0.7 \times HQ1$ as a starting point for initiating drawdown operations.

### 3.3. Flushing of the Existing Delta Formation at the Head of a Reservoir

Two separate delta flushing test runs were carried out. They are referred to as E1 and E2, respectively. They are based on the delta formation tests runs D3 and D2 as described in [9]. For each test run the experimental conditions were the same.

### 3.3.1. Water and Bed Surface Evolution during Reservoir Flushing

For both test runs, time dependent WSL changes (Δ WSL) at five different stations (X) along the experimental section are shown in Figure 5. In respect to the initial experimental procedure the flow conditions at time zero ($t = 0$) can be assumed stationary. Therefore, the decrease in the WSL indicates BSL erosion. For E1 and E2 the Δ WSL over time differ strongly along the flow path (X). One can distinguish a nonlinear trend between the downstream and the upstream end of the experimental section. At the end of the experimental section (X = 8.5 m) a strong decrease in height was measured during the first half hour (about 0.028 m for $t \sim 0.5$ h), while it remained nearly the same at X = 2.5 m. With the passing of time, the decrease in the WSL evolves further upstream until it reaches a constant after about 2–3 h. The varying drawdown patterns of E1 and E2 during t = 0.5 h and $t = 2$ h are a result from the different amount of measurements and time steps as well as the different initial size of the delta formations.

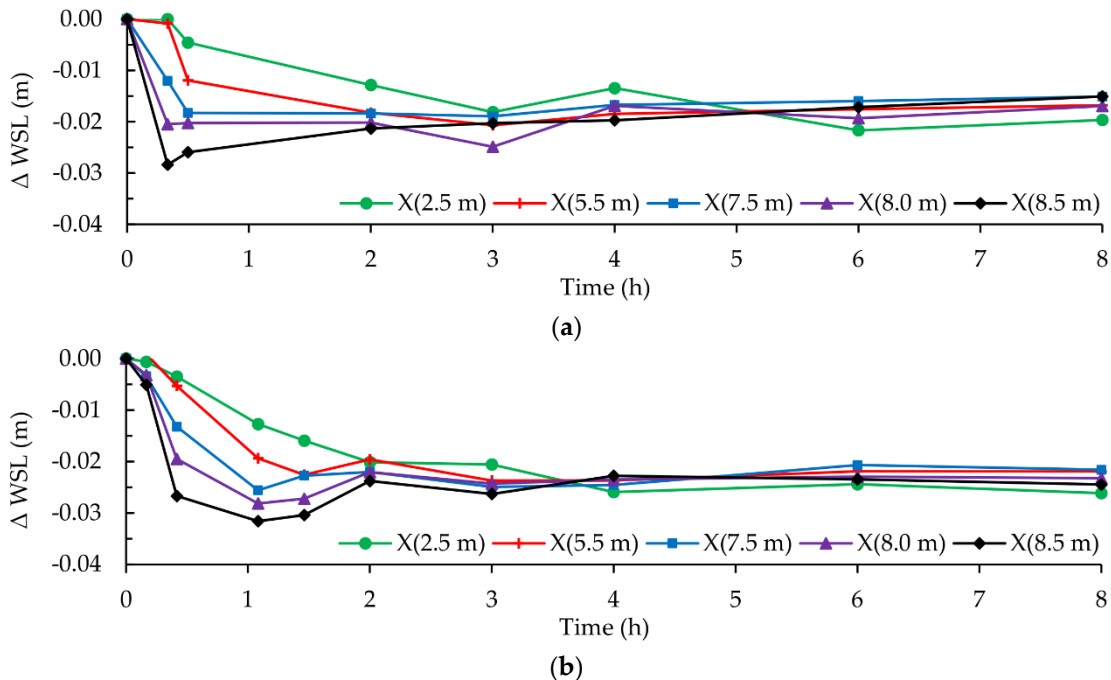

**Figure 5.** Water surface changes over the experimental time for different stations (X): (**a**) test run E1; (**b**) test run E2.

Figure 6 shows the evolution of longitudinal water and bed surface profiles for test run E1. The initial bed surface (black line in Figure 6) represents the result of the delta formation test run D3 described in [9]. The D3 experiment focused on the investigation of the upstream delta formation process. Hence, the delta front moved out of the downstream end of the experimental section. This led to an evenly bed and water surface level increase (≈0.02 m) which was almost parallel to the riverbed of the initial free-flowing section.

The WSL at the beginning of E1 (0 h) is illustrated as thick black line in Figure 6. A total of 20 min after the beginning of the drawdown the water levels remained unchanged for the section from X = 0.5–X = 6.5 m. In contrast, the water surface slope rapidly increased in the lower model section from X = 6.5–X = 8.5 m. This WSL profile is illustrated by the purple line in Figure 6. This indicates the beginning of the delta erosion process on the downstream end. The lowered water surface elevation at this section (X = 6.5–X = 8.5 m) can be explained by the rise of the slope, as well as a lowering of the bed surface due to the high local erosion rates. After 30 min the WSL profile (dark blue line) indicates that the drawdown has moved further upstream at a lower slope. After 2 h the bed and water surface slopes were parallel (light blue lines in Figure 6) with a similar decrease in elevation

compared to the initial bed and water profiles (−0.017 m). This erosion process continues but is almost negligible during the next two experimental hours. In the lower section the BSL of 2 and 4 h stay nearly the same while in the upper part the BSL after 4 h (green line) shows a lowering. The WSL behaves in the same way. This leads to the assumption that the water and bed surface slopes tend to be equal. Comparing the decrease in elevation, the delta erosion process can be described as highly nonlinear. During the experimental hours 4–6 and 6–8, Figure 6 implicates no significant changes regarding the WSL and BSL.

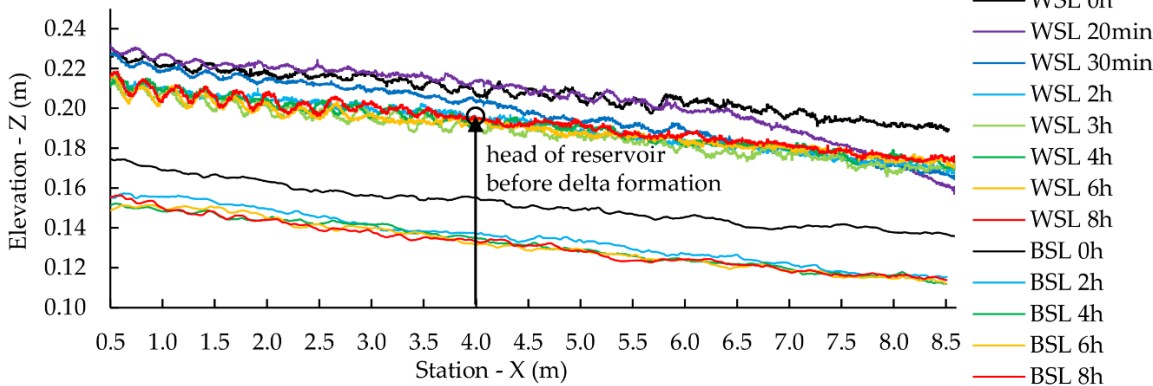

**Figure 6.** Bed and water surface profiles for test run E1 at the head of the reservoir at a flow rate of $0.7 \times HQ_1$ during flushing. WSL = water surface levels, BSL = bed surface levels.

The initial test conditions for the second test run E2 were provided by the delta formation test run D2 as reported by [9]. They constitute the riverbed and water surface levels after an 8-h lasting delta aggregation process. The delta front did not pass the downstream end of the experimental section. For this reason, the aggregated volume within the physical model is larger, compared to the initial conditions of E1. The initial WSL (0 h) representing the starting of the experiment is only available from X = 0.5–X = 4.0 m. For the downstream part the initial WSL was extrapolated by means of a linear regression line (thin black line in Figure 7).

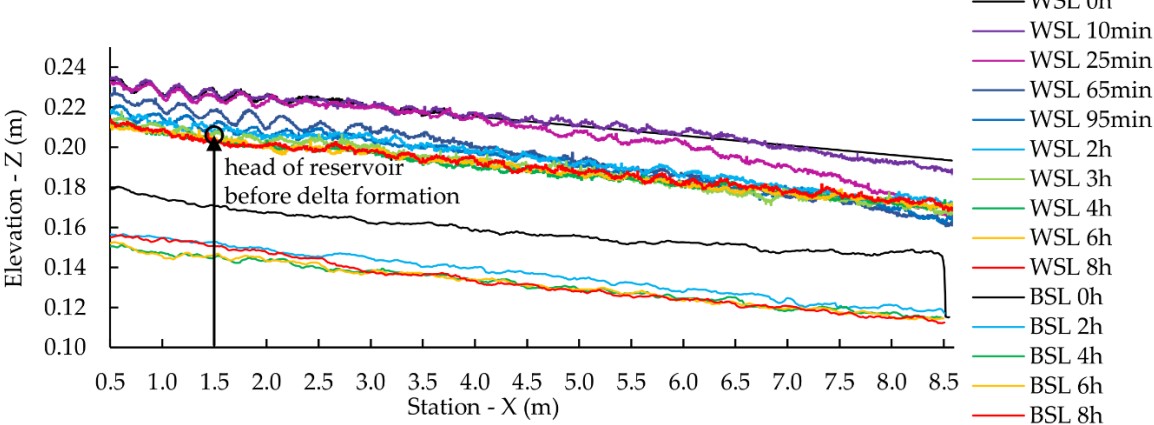

**Figure 7.** Bed and water surface profiles for test run E2 at the head of the reservoir at a flow rate of $0.7 \times HQ_1$ during flushing. WSL = water surface levels, BSL = bed surface levels.

Like in E1, the water surface slope in test run E2 started to increase in the lower model section. For the WSL after 10 min (purple line in Figure 7) the beginning of the drawdown is already apparent from station X = 7.0–X = 8.5 m, while the upper section (X = 0.5–X = 7.0 m) remains at the initial WSL. The WSL recorded at 25 min (magenta line) shows this backward process more distinct with a strongly bent drawdown curve from X = 6.0–X = 8.5 m and a less curved section from X = 4.2–X = 6.0 m.

The section above X = 4.25 m stays unaltered. The next WSL recorded at the 65 min (dark blue line) time step indicates that the drawdown curve has moved further upwards, reaching the upstream end of the experimental section. This WSL shows an elevation decrease over the total experimental section, with a linear trend from X = 4.2–X = 8.5 m. At the downstream end the WSL stayed on the same level between 25 and 65 min. At 95 min (blue line) the WSL gradient is constant from the upper to the lower end of the model also showing a flattened slope and a decrease in height compared to the previous measurement. This progress continues till the end of the second experimental hour. Comparing the 2-h BSL and WSL a similar extent of decrease in elevation is indicated (−0.018 m). The slopes differ slightly, with a steeper water surface slope of 0.56% and the corresponding bed surface slope of 0.5%. For the next two experimental hours (2–4), the erosion process continues. In the downstream section of the model, the BSL changes between hours 2 and 4 are less than in the upstream section, leading to a flatter slope of the BSL after 4 h (green line). The WSL undergoes the same progress. Considering the observed alterations, the delta degradation can be described as nonlinear and backward evolving. From testing hours 4–6 no change in the WSL and BSL is visible. At the end of the test run (8 h) the BSL shows a slight increase in height in the upstream model section from X = 0.5–X = 2.7 m. The magnitude of this local variation is in the range of the multiple observed bed level fluctuations of the free-flowing section. This indicates that the initial conditions of the free-flowing section were reached after 8 h.

Even though the two test runs E1 and E2 differ in terms of their initial extent, they are comparable in terms of the erosion process. After 8 h of experimental flushing, for both experiments the bed and water slopes are close to the dynamic equilibrium slope S0 of the free-flowing section. The mean BSL and WSL slopes can be found in the Appendix A section for both test runs (Tables A1 and A2). The WSL fluctuations can be explained by the uneven bed structure including sediment transport as dunes and flood flow. Due to the vertical exaggeration in Figures 6 and 7 the fluctuations appear even more pronounced.

### 3.3.2. Sediment Transport Rates during Reservoir Flushing

The transport rates were measured every hour for a constant flow rate of $0.7 \times HQ_1$. They show a strongly nonlinear decrease over time towards the preliminary measured transport rates of the free-flowing section presented in [9]. The difference in the transport rates of the experiments E1 and E2 can be explained by the smaller initial delta volume of D3. In Table 3 the amount of fed sediment as well as the measured transport rates during both test runs are summarized.

**Table 3.** Fed sediment load (input) vs. entrained sediment load (output).

| Test Hour (h) | Input E1 and E2 (kg) | Output E1 (kg) | Output E2 (kg) |
|:---:|:---:|:---:|:---:|
| 1 | 52 | 254 | 294 |
| 2 | 52 | 112 | 146 |
| 3 | 52 | 89 | 94 |
| 4 | 52 | 75 | 85 |
| 5 | 52 | 67 | 60 |
| 6 | 52 | 60 | 59 |
| 7 | 52 | 57 | 57 |
| 8 | 52 | 52 | 52 |

The changing transport rates of both experiments show a similar pattern, with a strong maximum in the first hour. During the second experimental hour the sediment load is less than half of the first hour. Over time the measured transport rates further converged towards the input rate which corresponds the equilibrium transport rate of the free-flowing section described in [9].

### 3.3.3. Measured Transport Rates vs. Volumetric Changes of the Photogrammetry

The calculated volumes changes based on the DEMs (Section 2.4) were converted into mass transport rates by means of the determined sediment bulk density and the knowledge of the sediment input and output. The volumetric changes were computed for an interval of 2 h, over the total extent of the experimental section (8.5 × 1 m). In Figure 8a the cumulative sediment transport rates are compared with the results of the volumetric analysis. The results of both measuring methods are nearly identical for E1 and E2. This underlines the accuracy of both methods and the reliability of the measurements. The high transport rates at the beginning of the flushing show a significant decrease during the first half of the experiment. During the second half the transport rates slowly merge into a constant value. During the first 2 h the erosion of the delta was predominant. The mobilized sediment load was nearly four times higher than the preliminary observed equilibrium conditions in the free-flowing section presented in [9]. Further, the influence of the sediment availability on the transport rates is shown in Figure 8a. Despite the different size of the delta volumes, E1 and E2 show comparable high transport rates during the first hour. In the next testing hours, the deviation of the two curves gets bigger, as in E1 the delta is already flushed out and therefore the sediment availability decreased. Figure 8b provides the delta erosion rates based on (i) the measured transport rates and (ii) the volumetric analysis. Again, the results of the hydrostatic weighing and the volumetric analysis have the same trend. According to the measured delta erosion based on hydrostatic weighing, both experiments show a rate of about 60% after the first hour. After 2 h the erosion rates sum up to 89% for E1 and 80% for E2. A flushing period of 4 h led to the total erosion of the delta, which was twice as fast as the time required for the delta formation.

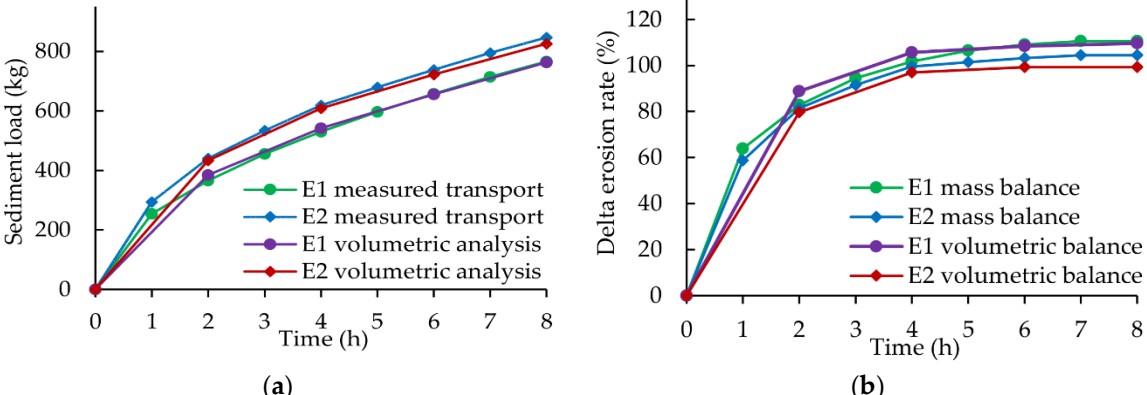

**Figure 8.** Measured transport rates and results from volumetric analysis, (**a**) cumulative sediment load; (**b**) delta erosion rate.

### 3.3.4. Retrogressive Erosion

Spatial and temporal differences in the erosion process were investigated by a volumetric analysis, based on the produced DEMs. Assuming a correlation between delta aggregation and delta degradation, the delta formation process was also analyzed in detail. The DEMs of different timesteps were subtracted to get a raster data set of the elevation changes. The results of the raster calculation can be found in the Appendix A section, named Figure A1 for the test runs D3 and E1. Figure A2 illustrates the computation outcome of D2, respectively E2.

Table 4 contains the mean elevation changes, for the delta formation test run D3 and the ensuing delta degradation E1 for different timesteps split into six 1 m long sections (S0–S5) as described in Section 2.4. The 8-h lasting delta aggregation indicates rising elevation changes along the flow direction X. This was expected, as the flow depth increases along the flow path, due to the backwater effect. The erosion E1 shows the same trend, but with a decrease in elevation. After 2 h (E1 0–2 h) the two downstream sections (S4 and S5) show similar absolute elevation changes as the corresponding

delta formation (D3). For this time step, from S1 to S3 the erosion rates were smaller compared to the delta aggregation. Considering the total experimental section, the main part of the delta was already eroded within 2 h of flushing. Still, some parts of the delta accumulation remain inside the model, mostly in the upper part. After 4 h (E1 0–4 h) a degradation of the riverbed below the initial level was observed. Therefore, the flushing of the delta aggregation was completed in less than 4 h, indicating a strong spatial and temporal nonlinearity. The following four testing hours led to an ongoing erosion of the initial riverbed, with a decrease in its intensity.

**Table 4.** Mean elevation change for 1-m long sections along the flow path for the test runs D1 and E1.

| Experiment | Time Step (h) | S0 (m) | S1 (m) | S2 (m) | S3 (m) | S4 (m) | S5 (m) |
|---|---|---|---|---|---|---|---|
| D3 | 0–8 | 0.015 | 0.018 | 0.017 | 0.019 | 0.019 | 0.022 |
| E1 | 0–2 | −0.015 | −0.017 | −0.016 | −0.018 | −0.019 | −0.022 |
| E1 | 0–4 | −0.015 | −0.019 | −0.020 | −0.022 | −0.021 | −0.023 |
| E1 | 0–6 | −0.017 | −0.021 | −0.021 | −0.021 | −0.022 | −0.024 |
| E1 | 0–8 | −0.019 | −0.020 | −0.021 | −0.022 | −0.022 | −0.023 |

Between testing hours 4 and 6, the dominant erosion took place in upstream part (S0 to S2). Further, during the last 2 h (6–8 h) a significant decrease in height was only measured for the section S0. This can be explained by natural height fluctuations due to the sediment transport as dunes. The downstream sections S1 to S5 show both decrease and increase in height. The backward evolving erosion process of the delta degradation seems to be completed after 8 h. The mean elevation changes for D2 and E2 are provided by Table 5. As expected, for the experiment D2 the delta aggregation shows rising elevation changes along the flow direction. The change in elevation for D2 shows higher values than for D3 due to the different location of the head of reservoir and therefore different magnitudes of the backwater effect.

**Table 5.** Mean elevation change for 1-m long sections along the flow path for the test runs D2 and E2.

| Experiment | Time Step (h) | S0 (m) | S1 (m) | S2 (m) | S3 (m) | S4 (m) | S5 (m) |
|---|---|---|---|---|---|---|---|
| D2 | 0–8 | 0.026 | 0.025 | 0.027 | 0.029 | 0.030 | 0.032 |
| E2 | 0–2 | −0.019 | −0.020 | −0.020 | −0.023 | −0.025 | −0.027 |
| E2 | 0–4 | −0.025 | −0.025 | −0.025 | −0.027 | −0.028 | −0.030 |
| E2 | 0–6 | −0.026 | −0.025 | −0.026 | −0.028 | −0.028 | −0.031 |
| E2 | 0–8 | −0.025 | −0.025 | −0.026 | −0.028 | −0.029 | −0.033 |

The first 2 h (0–2 h) of the flushing experiment E2 show a lowering of the BSL for all sections ($S_i$). The magnitude of the erosion revealed a significant trend along the flow direction X. For S0 the elevation decreased about 0.0191 m which represents 72% of the delta aggregation for S0. In comparison, the lowering of 0.0273 m at S5 represents 86% of the corresponding increase during the 8-h lasting delta formation. This indicates that the erosion process is backward evolving. For the total section, the mean erosion rate of the delta amounted 80% in this time step. After 4 h (0–4 h) 95% of the delta aggregation were eroded, with small variations along X. During the next two testing hours (4–6 h) the erosion of the delta sums up to be 97% showing no change in its characteristics. After 8 h of flushing, the total erosion rates of the delta inside the observed sections were 98%. Between the last 2 h (6–8 h) S0 shows a slight increase in height, which can be explained by natural fluctuations of wandering sediment dunes. All the other values show an ongoing erosion of the riverbed. For D2 the change in elevation was higher compared to D3 due to the different location of the head of reservoir and therefore different impacts of the backwater effect. This led to different erosion heights between E1 and E2.

In Figure 9a,c the mean section-wise elevation changes are illustrated. To make the test runs D3; E1 and D2; E2 comparable the different volume changes ($\Delta V_i$) were normalized by Equation (1). Trends in the normalized data were analyzed statistically by means of linear regression. Figure 9b

shows the normalized volume changes for the delta evolution for the experiments D3 and D2 and the linear model fitted to the data. The regression model (lmod.D3D2) shows an increase in volume growth of 5.2% per meter compared to the total volume change per timestep.

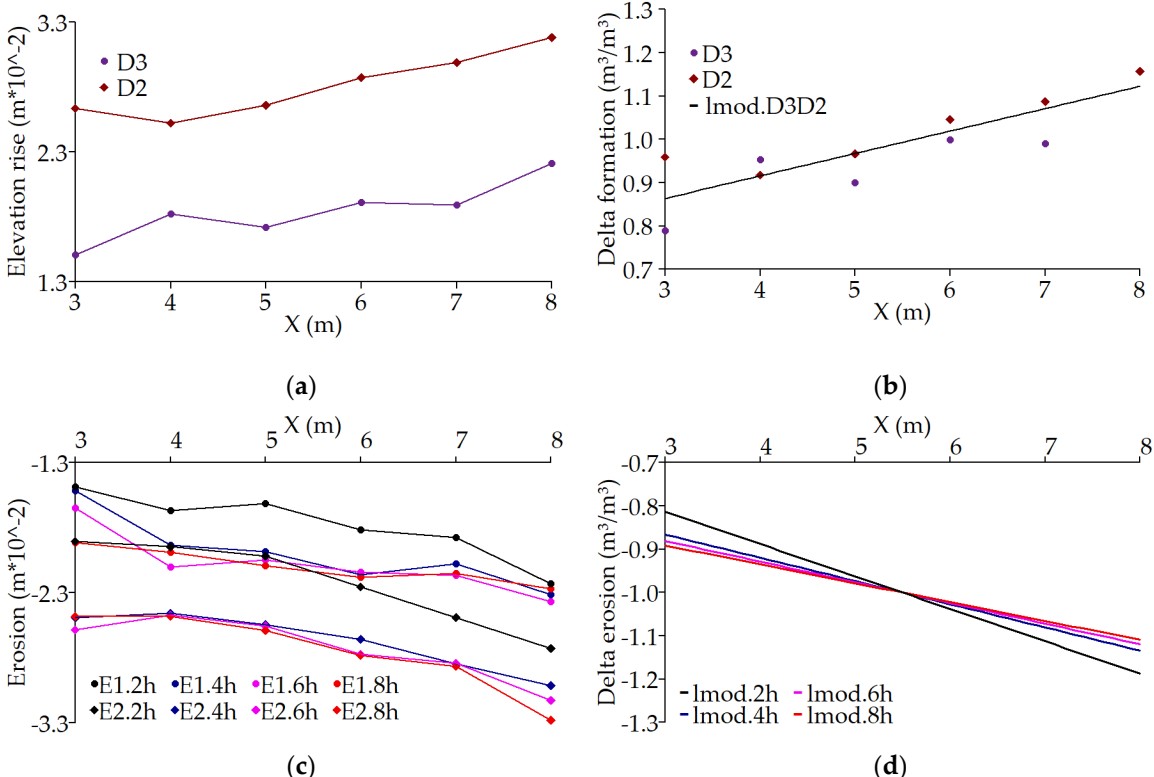

**Figure 9.** Delta formation and flushing processes as a function of the flow path (x): (**a**) delta elevation rise; (**b**) normalized delta volumes changes per meter in relation to the mean volume change during delta formation; (**c**) delta erosion during flushing experiments for different time durations; (**d**) linear regression models of the normalized erosion volumes changes per meter.

As shown in Figure 9d the slope of the 2 h flushing regression model (lmod.2h) is 1.5 times higher than the reverse one describing the delta evolution after 8 h (lmod.D3D2, Figure 9b). The models concerning the flushing period of 4, 6, and 8 h (lmod.4; lmod.6h; lmod.8h) show slopes and intercepts comparable to the reverse delta formation model as the erosion rate decreases. Hence, after about 4 h of flushing the riverbed conditions come close to the initial riverbed before the delta accumulated. Since lmod.2h is the only model which significantly differs from the others, the strong nonlinearity of the erosion process becomes evident. The statistical trend analysis indicates a retrogressive erosion process with its maximum during the first two experimental hours.

The parameters of the regression models and the resulting $R^2$ values are summarized in Table 6. All the model parameters indicate a high significance level ≤0.001. The $R^2$ exceeds 0.75 for all models representing an adequate description of the measurements. The lower $R^2$ values can be explained by the sediment transport as dunes. The dunes had a mean measured height of about 0.01 m along the free-flowing section. For an equilibrium slope of 0.005 a dune height of 0.01 m explains a local variation up to ±10% in relation to the mean bed elevation.

**Table 6.** Parameters of the combined linear regression models.

| Model | Period (h) | Slope [1] | Intercept [1] | $R^2$ |
|-------|-----------|-----------|---------------|-------|
| D3D2 | 0–8 | 0.052 | 0.708 | 0.752 |
| E1E2 | 0–2 | −0.075 | −0.588 | 0.900 |
| E1E2 | 0–4 | −0.054 | −0.705 | 0.796 |
| E1E2 | 0–6 | −0.048 | −0.737 | 0.769 |
| E1E2 | 0–8 | −0.044 | −0.760 | 0.828 |

[1] All model parameters show a significance level <0.001.

## 4. Discussion

The knowledge of the mean flood reoccurrence and the corresponding sediment transport rates is a prerequisite for the implementation of sustainable and successful sediment management strategies for hydropower. Therefore, it is useful to distinguish between flood events where gate operations are necessary from a flood risk perspective and events where gate operations are not required for flood prevention but where open gates could avoid reservoir sedimentation [9]. The findings show the importance of gate operations for discharges in the range of $0.7 \times HQ_1$—$HQ_1$ because of their significant contribution to the mean annual RSR. Statistically, for the Mürz river gauging station Kapfenberg-Diemlach this would lead to about four drawdowns per year due to flow rates between $0.7 \times HQ_1$—$HQ_1$ and additional two for events exceeding $HQ_1$. For events $>0.6 \times HQ_1$, the mean reoccurrence is 10 drawdowns per year, considering the period 1971–2016.

From field studies done by Harb et al. [19] and Badura [10] a data set comparing annual RSR to the corresponding C/I-ratios for many HPP all over Austria was provided (Figure 10). The field data show the strong correlation of rising annual RSR with decreasing C/I-ratios, best described by an exponential regression model (black line in Figure 10). In addition, the estimated annual RSR of the Mürz_model is displayed (green triangle).

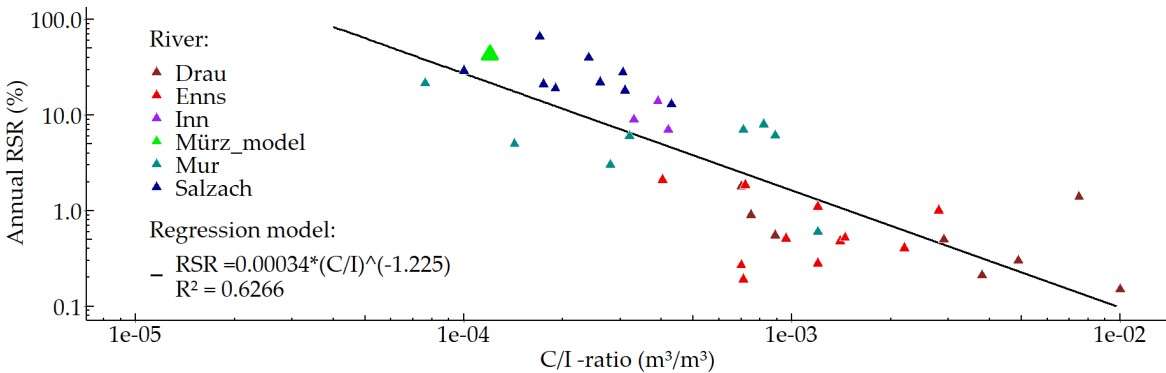

**Figure 10.** Reservoir sedimentation in Austria, field data and model results. Field data provided by [10,19].

The estimated annual 43% RSR of the Mürz_model fits into the data structure. For example, the Salzach HPPs show RSR values in the range from 13% up to 66% with a mean of 28% and a standard deviation of 16%. Based on the model simplifications and the assumption of full sediment availability, the Mürz_model represents a realistic scenario regarding the RSR of a medium sized gravel bed river.

To discuss the applicability of the model results regarding the implementation of sediment management strategies, a comparison to existing RoR HPPs was made, since some cases are already known for an improved flushing concept. Exemplarily, a newly constructed RoR HPP considering sluicing as well as flushing operations in terms of a sustainable reservoir management is the joint venture RoR HPP Gries, located at the Salzach river in Austria. With a design flow of 115 m$^3$/s, a hydraulic head of 8.9 m, and an installed capacity of about 8.85 MW, it is on the transition between

a small or medium sized HPP. Sediment connectivity aspects were considered in the structural planning as well as in the operation concept. Unlike most other RoR HPPs, the HPP Gries consists of two powerhouses, one each on the left and on the right riverbank, with two radial gates in between. The gates form a spillway aligned with the river axis [20]. Another feature of the Gries HPP is the low concrete weir height, since this is crucial to reduce the dead storage volume as described in [9,16]. Drawdowns are planned to be initiated for discharges in the range from $(0.63–0.86) \times HQ_1$ based on a newly erected gauging station combined with flood forecasting. The operators expect 10–20 drawdown events per year [21].

Compared to the Mürz model in Kapfenberg/Diemlach, the mean flow at the Gries HPP is 2.6 times higher, the $HQ_1$ is 2.4 times higher, the $HQ_5$ is 2.3 times higher, and the $HQ_{10}$ is 2.25 times higher. Nevertheless, the starting interval for a drawdown is in a similar range. Another example is the small or medium sized RoR HPP Bodendorf located in the Mur river in Austria. It has an installed capacity of 7 MW and a mean flow rate of 33 $m^3/s$ [10]. For Bodendorf, Schneider et al. [22] found values above $>0.62 \times HQ_1$ to be suitable for the initiation of drawdown operations. In terms of the annual RSR these RoR HPPs differ as this is strongly influenced by many local factors, such as (i) sediment availability, (ii) reservoir geometry and size, and (iii) the interaction with other HPPs (chain of HPPs) [9]. Still, the value of $0.7 \times HQ_1$ is stated to be a good reference point for the initiating of flushing events, as it seems to be applicable for a wide scope of RoR HPPs regarding the installed capacity.

With a gross head of 4.6 m, a resulting reservoir length of 1500 m and a design flow of 35 $m^3/s$, a maximum capacity of 1.4 MW can be calculated for the RoR HPP represented by the Mürz_Model. Compared to the installed capacity of the HPP Bodendorf and the HPP Gries, this value is about 5–6 times smaller. Considering all HPP located in Austria, approximately 87% do not exceed this capacity. Therefore, the Mürz_Model represents a wide range of RoR HPPs.

In this study drawdown operations at a RoR HPP headwater section were investigated at a flow rate of $0.7 \times HQ_1$. The findings show the suitability of low flood events for reservoir flushing if a drawdown is performed. The flood rate was capable of remobilizing an existing delta formation, built under the same flow rate without a drawdown. In addition, the entering sediment load from the free-flowing section upstream was also sluiced downstream. Thereby, the flow rate of $0.7 \times HQ_1$ was able to mobilize all grain size classes used in this study (14–120 mm in 1:1 scale).

In Sindelar et al. [9] the reservoir sedimentation process at the headwater section of the same RoR HPP was simulated with the water levels kept at operation level and a discharge of $0.7 \times HQ_1$. The findings revealed that under these conditions the bed load transport stops for grain sizes >14 mm at a 1:1 scale. Almost all sediments settle in the reservoir head section resulting in a rise of bed and water levels. This rise will not increase flood risk for the simulated event. However, these events show a high reoccurrence probability and will therefore lead to a further rise of the delta and the water levels if the reservoir is not drawn down. For a subsequent high flood events this could lead to a significant increased flood risk [9].

With the beginning of the drawdown, the sediment transport capacity increased rapidly due to the high energy slope and therefore high shear stresses. The maximal erosion occurred at the downstream end of the deposition. The erosion continuously evolved upstream, causing the steep slope at the former delta front to decrease. The slope at the delta top and tail increased until both slopes equalized. This indicated the end of the retrogressive erosion as described by [14]. For the water surface evolution, the same trend was observed. This also matches the findings of [14]. Hence, the increased flood risk caused by deltaic depositions was diminished. Figure 11 shows a scheme of the retrogressive delta erosion process for equal time steps, each illustrated by a different color. After reaching free-flowing conditions the erosion process was initiated at the downstream end of the delta deposition caused by the high energy slope at the delta front. Starting with high transport rates, the sediment concentration decreased for each timestep, which is also illustrated by the graph of the sediment concentration at the downstream end of the scheme. The erosion scheme also illustrates the water level decrease based on the experimental observations.

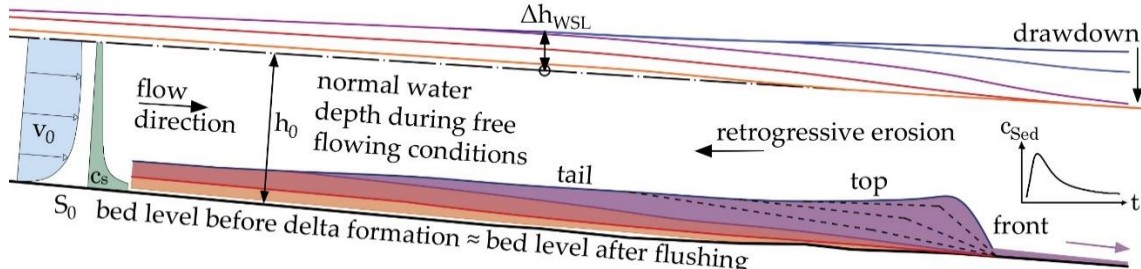

**Figure 11.** Spatiotemporal scheme of the retrogressive delta erosion located at the head of a run-of-river reservoir. Dashed black lines inside the delta deposition indicate the beginning of the retrogressive erosion.

According to [14] there is no dependency on any specific grain sizes for the appearance of a retrogressive erosion, still the characteristics and possible erosion patterns are influenced by the grain size distribution of the deposits. For the Hengshan storage reservoir in China the retrogressive erosion of unconsolidated silt material took less than 24 h for a 2000 m long section [14].

Additionally, retrogressive erosion was observed and investigated for a coarse river delta by Randle and Lyon [23], at the Elwha river in Washington [14]. This is in line with the findings of the present paper, as presented in the retrogressive erosion scheme of Figure 11. For the Sanmenxia reservoir at the Yellow River rapid full drawdown operations led to retrogressive erosion accompanied by high amounts of mobilized sediments. This caused serious depositions and increased flood risk in the downstream river section [24].

When the delta front started to degrade, the bed load transport rate was nearly four times higher than the corresponding value during free-flowing equilibrium conditions. In total, the delta was remobilized twice as fast, as it aggregated in the first place. Over time the sediment transport decelerated as the delta formation was mostly eroded and the sediment availability converged towards the initial equilibrium conditions. This points out the suitability of drawdowns, even for short periods under the present conditions. Together with the results of Sindelar et al. [9,25] and the present findings, a resettling of the eroded material is expected for partial drawdowns as the backwater effects reoccur downstream. Still, partial drawdowns can be suitable at the headwater section during flood events of low intensity to (i) remobilize deposited sediments and (ii) sluice the incoming bedload further downstream within the reservoir. Combining more frequent partial drawdowns with ensuing full drawdown flushing may result in a good outcome. This strategy may be described as a combination of reservoir head sluicing and near weir flushing. This means that flood events with low intensities are used to sluice the incoming sediments downstream into the impoundment. For events with high intensity a full drawdown is initiated to remobilize the deposited material. As the subsequent filling of the reservoir is less time consuming, partial drawdowns reduce the energy revenue losses. This fact could be relevant for RoR HPPs with high C/I ratios. The energy costs may further decrease, as the turbine may be kept in operation [19]. To provide evidence, further investigations of sections in the vicinity of the weir are necessary.

Discussing the consequences of these findings for future hydropower operations, the following key aspects are worth considering: (i) ecology, (ii) flood risk, (iii) technical issues and energy revenue. In terms of ecology and river morphology the conservation of sediment connectivity should be pursued. This is mostly achievable if reservoirs are drawn down in periods of high sediment transport, enabling the sediments to pass the impoundment. Further, the design of the HPP shows significant influence on the success of flushing operations. Especially, the effects of fixed weir heights and reservoir widenings must be considered [16].

From a flood risk perspective, aggregations in the headwater section have the potential to increase flood risk as they raise the bed levels and therefore reduce the effective cross section. Considering the impoundment as a component of the HPP-facility, reservoir sedimentation can cause operational

problems due to blocking of the turbine inlet or outlet [19]. Resulting maintenance work is cost intensive while simultaneously restraining the energy production. Further, possible loss in gross head can reduce the energy revenue. In contrast, drawdown operations also potentially harm the environment, if there is no adequate concept. High additional suspended sediment load, increased bed load transport, and the release of organic material provoke high environmental stress, such as critical oxygen consumption [15]. Further, drawdown operations lead to additional costs and energy revenue losses. The attempt to bring all these issues in line, highlights the challenges future hydropower development must face to keep its key role in the energy transition.

## 5. Conclusions

Within the present paper the consequences of drawdown flushing operations on delta formations were investigated at the headwater section of a RoR HPP. Retrogressive erosion processes were found to be dominant during the initial flushing phase, accompanied by high sediment transport rates, leading to a remobilization of the existing delta twice as fast as the preliminary formation process took. These findings confirm the suitability of low flood flows for reservoir flushing operations. In terms of ecology, flood risk, technical, and economical reasons the value of $0.7 \times HQ_1$ might state a good reference point for the initiation of a reservoir drawdown. Even short drawdown periods are capable of re-mobilizing a respectable amount of existing delta depositions. For partial drawdowns, the bed load material is expected to resettle more downstream within the impoundment. Field measurements at many Austrian RoR HPPs and the present physical model results indicate the need for sustainable sediment management strategies, to ensure sediment connectivity and avoid bed erosion and land loss in the downstream river reaches. Therefore, more systematic investigations regarding the vicinity of the weir including gate operation strategies should be pursued.

**Author Contributions:** Conceptualization, K.R.; methodology, K.R., T.G.; formal analysis, K.R., C.S., T.G.; investigation, K.R.; T.G.; writing—original draft preparation, K.R., T.G.; writing—review and editing, C.S., H.H., C.H.; visualization, T.G., K.R.; supervision, C.S.; H.H.; C.H., funding acquisition, C.H.; H.H.; K.R. All authors have read and agreed to the published version of the manuscript.

**Funding:** The financial support by the Austrian Federal Ministry for Digital and Economic Affairs, the National Foundation of Research, Technology and Development of Austria is gratefully acknowledged. We gratefully acknowledge financial support from the Christian Doppler Research Association.

**Acknowledgments:** We kindly acknowledge the support of Hannes Badura and Gabriele Harb for providing their field data.

**Conflicts of Interest:** The authors declare no conflict of interest. The funders had no role in the design of the study; in the collection, analyses, or interpretation of data; in the writing of the manuscript, or in the decision to publish the results.

## Appendix A

**Table A1.** Mean slopes of the measured bed surface levels and water surface levels for test run E1, n.d. = not determined.

| Time (hh:mm) | Bed Surface Slope (%) | Water Surface Slope (%) |
|:---:|:---:|:---:|
| 00:00 | 0.46 | 0.44 |
| 00:20 | n.d. | 0.77 |
| 00:30 | n.d. | 0.72 |
| 02:00 | 0.55 | 0.55 |
| 04:00 | 0.49 | 0.51 |
| 06:00 | 0.51 | 0.47 |
| 08:00 | 0.50 | 0.49 |

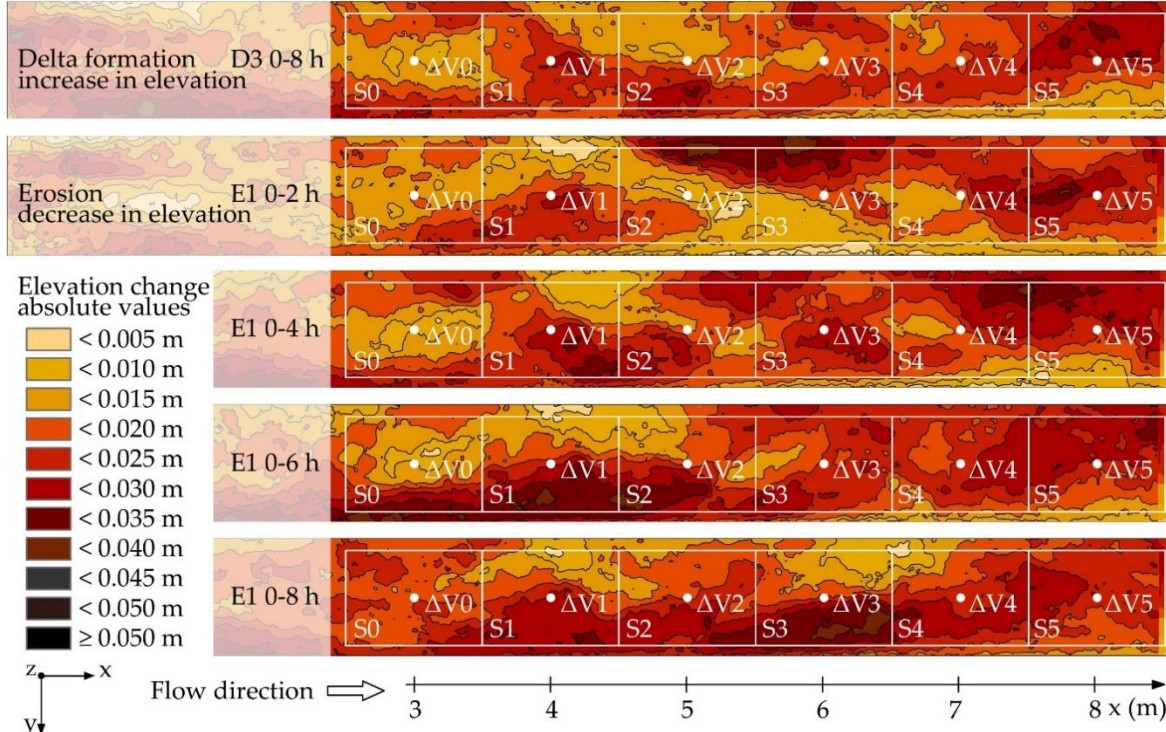

**Figure A1.** Volumetric analysis of the delta formation D3 and the ensuing erosion E1 during drawdown flushing.

**Table A2.** Mean slopes of the measured bed surface levels and water surface levels for test run E2, n.d. = not determined.

| Time (hh:mm) | Bed Surface Slope (%) | Water Surface Slope (%) |
|:---:|:---:|:---:|
| 00:00 | 0.42 | 0.48 |
| 00:10 | n.d. | 0.58 |
| 00:25 | n.d. | 0.72 |
| 00:65 | n.d. | 0.75 |
| 01:35 | n.d. | 0.66 |
| 02:00 | 0.50 | 0.56 |
| 03:00 | n.d. | 0.56 |
| 04:00 | 0.45 | 0.50 |
| 06:00 | 0.46 | 0.50 |
| 08:00 | 0.53 | 0.49 |

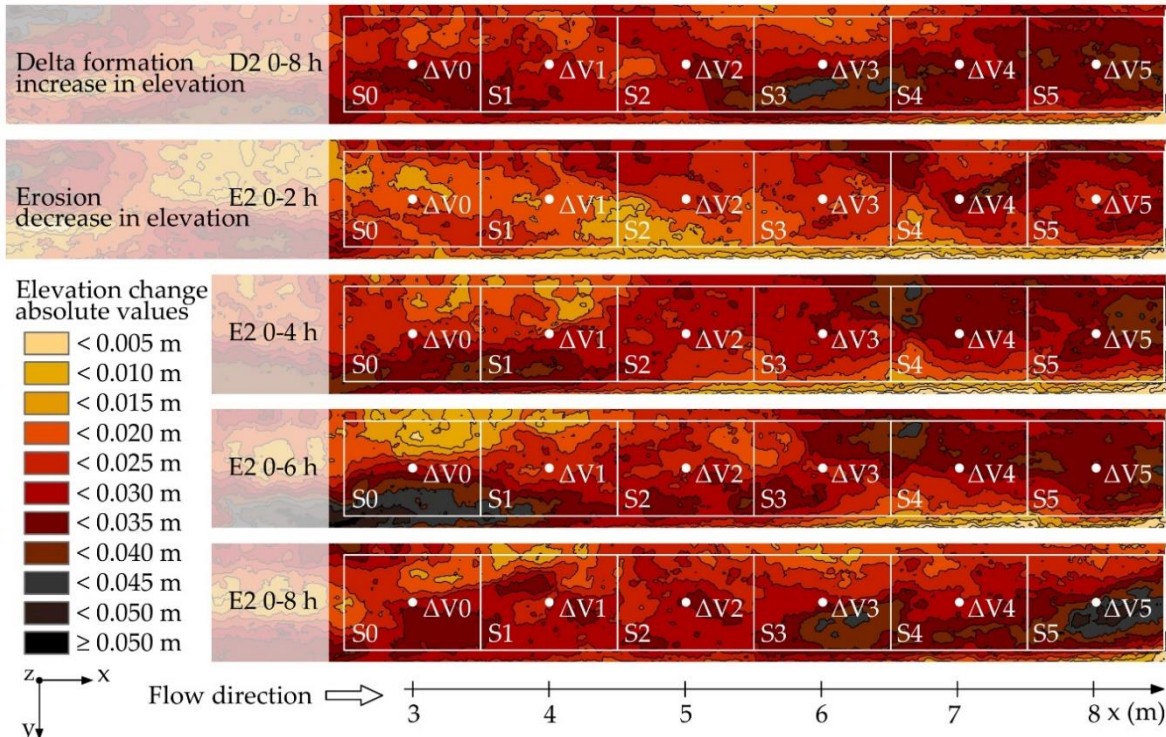

**Figure A2.** Volumetric analysis of the delta formation D2 and the ensuing erosion E2 during drawdown flushing.

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
