# Peer review of "Experimental Study at the Reservoir Head of Run-of-River Hydropower Plants in Gravel Bed Rivers. Part II: Effects of Reservoir Flushing on Delta Degradation"

_water, doi:10.3390/w12113038_

Round 1
Reviewer 1 Report
The authors submitted the manuscript entitled "Experimental Study at the Reservoir Head of Run-of- River Hydropower Plants in Gravel Bed Rivers. Part II: Effects of Reservoir flushing on Delta Degradation" for possible publication to Water. I was only reviewing this part and have no knowledge on part I. Having said this, I found that the paper can stand alone. The paper deals with a highly relevant topic and is, in general, well written. Nonetheless, I have a number of comments that should be addressed by the authors.
L15: "... representative river..." is a bit confusing (and why is it representative; later the river Mürz is mentioned)- this could be phrased more clearly.
L21: How was this time-scale determined for the 1:1 scale (see my comment at the end of the review).
L28: This is no criticism - I find the introduction is written nicely and that it provides a lot of useful information.
L102: This paragraph is not clear to me, as it seems to be based on results of the scale model which has not yet been introduced.
L104 and following: Why giving the units for the sediment transport model? It would be better to provide some more information on the model itself. What kind of information was available for cross-checking the accuracy of the model?
L110: How was the initial reservoir volume known?
L127: Can you please comment on how the draining of the flume may have affected the results?
L164: I take it from this sentence that the experiments were carried out with sediment feeding. This could be mentioned more explicitly.
L173: What was the width of the model (or the flume)?
L186: Equation 1 (and in the text) - it would be good to use subscript for the indices.
L197: Isn't this the background for the data provided in Table 1? The hydrological analysis is also not the main objective of the paper, so that this part may be relocated into section 2?
L222: The fitted tri-linear transport model remains a black-box to me (and how it was calibrated/validated).
L253: Remove brackets from (X = 2.5 m).
L266: Please check the figure - the legend is cut off in my pdf version.
L338: Here the flume width seems to be mentioned...(see my comment above).
L394: This must be a decrease of 0.019 m?
L409: This must be the data from reference [9]? Maybe it would be worth to describe the initial situation for D3 and D2 a bit more detailed?
L410: See my comment above regarding the "black-box model". Some more information would be helpful.
L433 and following: I like the dicussion, it is nicely written.
L21: I cross-checked the statement about the time scale given in the abstract when reading the manuscript but could not find any justification. It seems to me that the model-scale (1:20) was meant? If not, much more details should be given how the time was upscaled, as this is not straightforward.
Reviewer 2 Report
The comments are in the attached file
